# The effect of prolonged closed-loop management on athletes' sleep and mood during COVID-19 pandemic: Evidence from the 2022 Shanghai Omicron Wave

Chenhao Tan[1], Jinhao Wang[1], Jun Yin[1], Guohuan Cao[1], Lu Cao[2,3], Chao Chen[4], Jun Qiu[1] *

1 Shanghai Research Institute of Sports Science (Shanghai Anti-Doping Agency), Shanghai, China, 2 Basic Education Office, Tongji University, Shanghai, China, 3 School of Marxism, Tongji University, Shanghai, China, 4 No.1 High School Affiliated to Tongji University, Shanghai, China

* qiujun@shriss.cn

## Abstract

Closed-loop management of athletes at the training base is a compromise approach that balanced epidemic prevention and sports training during the COVID-19 pandemic. This study investigated the impact of prolonged closed-loop management on athletes' sleep and mood during the 2022 Shanghai Omicron wave. The Pittsburgh Sleep Quality Index and the Profile of Mood States were used to assess the sleep and mood states of 110 professional athletes in "closed-loop management" at the training base after 1 and 2 months of closed-loop management, respectively, to characterize changes in sleep and mood with prolonged closed-loop management. After two months of control, the sleep and mood of 69 athletes and students of the same age were measured using the Pittsburgh Sleep Quality Index and Perceptual Stress Scale, as well as the Warwick-Edinburgh Mental Well-being Scale, to compare the differences in sleep and mood between athletes undergoing closed-loop management and the general population who were managed in the community. Paired sample *t*-tests and independent sample *t*-tests were used for comparisons across different time intervals and different management approaches. Results showed that with the time of closed-loop management increased, athletes woke up earlier ($p = 0.002$), slept less ($p = 0.024$), and became angrier ($p = 0.014$); athletes had poorer overall sleep quality ($p < 0.001$) but lower stress level ($p = 0.004$) than those who were outside the base. In closed-loop management, the athletes were able to maintain a stable sleep and mood state. Sports team administrators must be aware of the need to improve athletes' sleep quality and help athletes to agree with this approach of management.

## Introduction

Because of its high infectiousness, the COVID-19 epidemic at the end of 2019 quickly swept through countries and became a pandemic. Facing such an outbreak, numerous governments

**Data Availability Statement:** All relevant data are within the manuscript and its Supporting information files.

**Funding:** This research was supported by the Shanghai "Science and Technology Innovation Action Plan" social development science and technology research projects (22dz1204601) by the Science and Technology Commission of Shanghai Municipality. The funders had no role in study design, data collection and analysis, decision to publish, or preparation of the manuscript.

**Competing interests:** The authors have declared that no competing interests exist.

and regions adopted a lockdown control strategy [1]. Lockdown strategies implemented effectively halted the spread of the virus and saved the lives and health of millions of people on time. Lockdown following the outbreak was demonstrated to have a positive effect on epidemic. However, there are two sides of each coin. Lockdown has gradually been discovered to have a significant detrimental impact not only on social level activities such as daily work, schooling, economic activities, and social connection but also on people's physical and mental health at the individual level [2,3]. It has even been viewed as another public health phenomenon, which has sparked some debate [2,4].

For competitive athletes, the issues of lockdown are considerable. Lockdown does not mean resting at home or working from home with a decreased commute for athletes in competitive sports, but rather creating the conditions to continue training at a certain level and without the opportunity to compete [5,6]. In such circumstances, they must remain physically and mentally prepared for future competitions. Sleep and emotional state are relatively major aspects influencing performance for athletes in lockdown [7,8]. However, research on the influence of lockdown on sleep and mood has repeatedly shown that it induces significant negative changes in sleep and mood [9–15]. This is especially true for athletes. Athletes, like the general public, exhibit comparable patterns of sleep and psychological changes during a lockdown, according to researchers. For example, one study of 399 Australian professional athletes discovered that under the influence of lockdown, athletes experienced an increase in sleep duration, delayed sleep time, and sleep latency [16]; Spanish handball players experienced reduced sleep quality and became emotionally negative during isolation [17]; a survey in 3911 athletes from 49 countries revealed an increase in sleep duration, delayed sleep time [8]; and it is especially noteworthy that a study conducted on 1454 elite athletes found worse sleep quality in lockdown and it also found that those with reduced training intensity had poorer sleep quality, which adds insult to injury [7]. Furthermore, some researchers tracked the mental states of 45 adolescent athletes at different stages of a 10-week lockdown and discovered that negative mental states were stronger in the later stages of the lockdown than in the earlier stages and that players in group sports were more severely affected [18].

It is essential to notice that, governments are actively adapting in attempts to reestablish a relatively normal order of life and work. This is also true in competitive sports. During the widespread lockdown at the beginning of the outbreak, the majority of athletes maintained a strong desire and needed to train, but only a small number of athletes were able to sustain adequate training [6]. Athletes were denied access to sufficient training environments and protections due to factors such as a lack of specialized facilities or venues, distant coaching, and a lack of medical and rehabilitation conditions [19]. Following the initial experience with lockdown, the game's needs spurred the development of other coping strategies. With significant events like the Tokyo Olympics and the Beijing Winter Olympics on the horizon, more and more sports administrators are focusing on how to balance epidemic prevention with competition and training goals. For example, in the crisis of the Omicron epidemic in early 2022, the organizing committee used closed-loop management to assure the safety of athletes participating in the Beijing Winter Olympics [20].

Aside from sports events, some people have applied closed-loop management to training. For example, researchers have reported a well-tested approach known as "quarantine" training camps (or "bubble" camps) that aims to provide a safe and professional training environment for athletes competing in the Tokyo Olympics [6,21]. A "bubble" camp can be defined as a quarantine-style camp whereby a group of people (e.g., athletes, support staff) is strategically isolated from wider society to resume regular activities for a specified period of time [22]. It is important to note that although researchers use different terms such as "quarantine camp", "bubble approach", and "closed-loop management", the concepts are essentially the same.

The closed-loop management training meets the requirement of epidemic prevention, protects athletes' health, and produces an appropriate environment for specific training; hence it is recommended as an effective training approach in epidemic prevention [6]. However, it needs to be realized that it is still a type of lockdown, and athletes' sleep and emotions may be affected as well. However, most existing research on the sleep and mood states of athletes under lockdown has concentrated on isolation at home rather than in a training environment, and research on the sleep and mood changes of athletes under closed-loop management arrangements is extremely limited. Only one practice, which was the first one to employ "quarantine" camps, has conducted a comparison. This study revealed that athletes' emotional well-being improved over a one-month training camp period, with no difference from the state before the home isolated period; sleep duration was reduced relative to the home isolated period, as were sleep problem behaviors, but this difference did not differ [21].

According to the findings of this study, closed-loop management has a positive effect on athletes' moods but a smaller effect on sleep issues. However, by the time the study was started, the athletes had already been exposed to a nationwide lockdown outside of the camp, which may have interfered with the impact of the camp itself. And, because of training camp in this study was used as part of a program in which everyone participated for no more than 30 days, the results have little or no way to reveal the characteristics of the effects of this method on mood and sleep over a longer period of time. However, longer closed-loop management is probable, particularly if more infectious variants emerge or before significant events. Finally, in terms of methodology, while the study asked athletes to describe their sleep and mood status before lockdown, during a lockdown, and during camp, these responses were collected simultaneously at the end of training camp. This strategy may have resulted in bias in the reporting.

Closed-loop management has emerged as a promising approach to strike a balance between epidemic prevention/control and training effectiveness. This study seeks to expand on previous research regarding the effects of one month of closed-loop management on sleep and mood among athletes. Specifically, the study aims to compare the impact of closed-loop management over a longer duration (approximately 2 months) both longitudinally (in comparison to the effects observed after approximately 1 month) and horizontally (in comparison to management in the community / home isolation) to comprehensively evaluate the efficacy of this management style. To achieve these objectives, the study undertook an analysis of the sleep and mood of professional athletes under closed-loop management at different time points (after approximately 1 month and after 2 months) during the Omicron Wave in Shanghai, China, 2022. Additionally, the study conducted a cross-sectional comparison between high school-aged athletes under closed-loop management and high school students of the same age who were in home isolation for approximately 2 months. High school students were chosen due to their requirement to follow a daily schedule of online classes, including physical education, while in isolation, thereby offering a comparable routine to that of the athletes. The study's findings have practical implications for promoting targeted sleep and mood precautions during future outbreaks or other public health emergencies.

## Materials and methods

### Participants

The subjects in this study consisted of three samples.

Sample 1 was used to compare the athletes' sleep and mood after 1 month and 2 months of closed-loop management. It comprised 249 athletes aged 16 years and above who trained at Shanghai Chongming Sports Training Base during the Omicron wave in Shanghai, China, 2022. All athletes were subjected to closed-loop management. The sample size represented the

vast majority of athletes in the targeted age range who trained at the facility under closed-loop management during the study period. However, not all athletes participated in both tests due to decisions made by administrators of each sports team on participation in each test. Specifically, 178 athletes participated in the test after 1 month of closure, while 181 athletes participated in the test after 2 months. Among these athletes, 110 participants (70 females) took part in both tests, providing longitudinal data for analysis. The average age of the participants was 19.42 years ($SD$ = 2.99). The sports included in the study were fencing, modern pentathlon, field hockey, Chinese martial arts, gymnastics, boxing, badminton, judo, taekwondo, and karate.

In this study, obtaining permission from sports team administrators was necessary for athletes to participate and for the use of their data. However, some athletes did not participate in either the first or second test due to sports team administrators failing to provide timely feedback on their intention to participate, or because they did not perceive the need for their sports team to participate. Notably, none of the athletes themselves expressed any desire to withdraw from the study.

Sample 2 and Sample 3 were used for comparison between closed-loop management and management in the community (home isolation).

Sample 2 consisted of 69 athletes aged 16–18 years who participated in the test after 2 months of closure management (43 females), with a mean age of 16.96 years ($SD$ = 0.83). The sports included fencing, modern pentathlon, field hockey, handball, Chinese martial arts, gymnastics, boxing, badminton, judo, taekwondo, and basketball.

Sample 3 consisted of 160 high school students (85 females), with a mean age of 16.10 years ($SD$ = 0.30). The students came from a senior high school in Shanghai. The students were required to attend online classes and interact with their teachers for the majority of the day, from Monday to Friday. They were also required to complete homework after class and participate in physical education classes that were included in the curriculum and needed to be completed at home. The schedule was set by the city, and the online classes were recorded and broadcast by the city. This daily routine is similar to that of the athlete's training and study routine in the training base. The students who were isolated at home did not have to deal directly with many of the material concerns of life, similar to the athletes who were training in the training base. None of the participants in Sample 2 had any prior experience of being placed in isolation in a cubicle hospital or isolation site. At the time of testing, all students had been studying at home for approximately 2 months.

The sample size was estimated by G*Power 3.1.9.2 software. According to the design of this study, for the analysis of sample 1, when using the paired samples $t$-test, the minimum sample size was calculated to be 54 (based on an effect size of 0.50, an α-level of 0.05, and a power of 0.95). There was a total of 229 subjects in Sample 2 and Sample 3. When using the independent samples $t$-test and set an effect size of 0.50, an α-level of 0.05, and a power of 0.80, the minimum sample size was calculated to be 128 (64 participants for each group).

The participants completed all questionnaires on an online survey site using their own cell phones (wjx.cn). To avoid missing answers, this website checked the responses before participants submitted the questionnaire. As a result, there were no missing data in this study. We set the following exclusion criteria: (1) participants who completed the questionnaire more than once in the same test; (2) participants who did not fill in their personal information truthfully (e.g., using a pseudonym); and (3) participants who did not fill out the PSQI correctly (e.g. filled in the same sleep time and wake time). No subjects were excluded from this study.

This study used data from the psychological services undertaken by the Institute during the lockdown. Data of athletes were obtained from routine psychological services for sports teams. Data of the students were obtained through the school's psychological health education

program implemented during the epidemic. The test used in this program was adapted from tests conducted in sports teams, with some minor adjustments made in accordance with the school administration's requirements. Following the tests, the school's psychology teachers provided mental health education based on the results. Ethical approval for the study was obtained from the Research Ethics Committee of Shanghai Research Institute of Sports Science (No. LLSC20220005). Written informed consent form were obtained from participants or their guardian. Participants who were unable to obtain paper forms because of lockdown used electronic signatures. All procedures were in accordance with the 1964 Helsinki declaration and its later amendments or comparable ethical standards.

## Measurements

**Pittsburgh Sleep Quality Index (PSQI).**   Sleep quality was assessed using the Chinese version of Pittsburgh Sleep Quality Index (PSQI). PSQI is a self-report instrument comprised of 19 items evaluating seven components of sleep: subjective sleep quality, sleep latency, sleep duration, sleep efficiency, sleep disturbances, daytime dysfunction, and use of sleep medications. The seven components can be added together to produce a global score ranging from 0 to 21. Participants rate their overall state in the previous month; higher scores indicate poorer sleep quality. The instrument exhibits adequate psychometric properties [23]. The Chinese version of the PSQI is also widely used in the Chinese population and has demonstrated high reliability and validity.

In this study, the scores of some dimensions were utilized individually as dependent variables in order to describe the characteristics of athletes' sleep in detail, with reference to the methods of previous studies [9,24]. Thus, we have the following indicators in this study: (1) global PSQI score, the total score (0–21) of PSQI, with higher scores indicating poorer sleep quality overall; (2) subjectively rated sleep quality, score on the subjective sleep quality dimension of the PSQI (0–3), with higher scores indicating worse subjective perceived sleep quality; (3) Sleep time, the time when participant go to bed (in minutes since 0:00, if later than 24:00, 1440 minutes are superimposed), the bigger the value, the later the bedtime; (4) Wake time, the time participant wake up after going to bed (in minutes since 0:00), the greater the value, the later the morning wake time; (5) Sleep latency, the time between going to bed and falling asleep, with larger values indicating that it takes more time to fall asleep; (6) hours of sleep, the total number of hours slept, with higher values indicating longer sleep duration; (7) calculated sleep efficiency, the proportion of total sleep time actually spent asleep after falling asleep, with higher values indicating higher sleep efficiency; (8) total sleep disorder score, the total score for the sleep disorder dimension of the PSQI (0–30), with higher scores indicating more severe sleep disorder symptoms (symptoms such as difficulty falling asleep, waking up at night, excessive dreaming, etc.); and (9) total daytime dysfunction score, the total score for the daytime dysfunction dimension of the PSQI (0–6), with higher values indicating more pronounced daytime dysfunction (feelings like sleepiness, problem in keep up enough enthusiasm, etc.).

In regards to sleep medication, Chinese athletes are required to obtain administrative approval for use exemptions under anti-doping regulations after receiving medical advice. Similarly, for students, the use of psychotropic substances is subject to information collection in schools for youth protection purposes, allowing for school psychologists to intervene and provide assistance when necessary. However, in this study, neither type of formal use of sleep medication was found. Furthermore, the scarcity of medical resources during the epidemic control period made seeking medical attention for sleep problems highly unlikely. As such, changes in the use of sleep medication can be largely disregarded in this study. To avoid

confounding the results with factors limiting medication use, the dimension of sleep medication was not included in the statistical analysis.

**Profile of Mood States (POMS).**   Athletes' mood was assessed by using the Profile of Mood States (POMS). The POMS is a reliable and valid measure of subjective mood states that has been used in numerous studies. The Chinese version of the POMS consists of 40 adjectives that define 7 mood states: tension (e.g. restless, nervous), anger (e.g. peeved, bitter), fatigue (e.g. weary, bushed), depression (e.g. hopeless, helpless), vigor (e.g. cheerful, energetic), confusion (e.g. bewildered, forgetful), and esteem (e.g. proud, satisfied). This study used the Chinese version that is widely used in China [25]. Participants rate their subjective feelings in the previous week on a five-point Likert-type scale ranging from 0 (not at all) to 4 (extremely). We used the total score of each dimension as the score of each emotion in this study.

**Perceptual Stress Scale (PSS-10).**   Perceived stress scale was used to assess the effect of home isolation on emotion, in reference to previous studies [13]. It has been validated as a measure of perceived stress among various populations and its score moderately correlates with anxiety and depression scale. The PSS-10 was found to measure felt stress levels better in a broader population in one research with a Chinese sample [26]. Hence the 10-item Perceived Stress Scale (PSS-10) was utilized in this study [27]. The PSS-10 is made up of six negative items (perceived distress, the lack of control and negative reactions) and four positive items (coping capacity, the degree of ability to cope with existing stressors), which are evaluated on a 5-point scale ranging from "none" to "always." Positive items were reverse-scored and added to negative items to obtain a total perceived distress score, with higher scores indicating greater perceived distress.

In the present study, the PSS-10 was employed to assess and compare mood experiences between athletes and students in Sample 2 and Sample 3. The substitution of the POMS with the PSS-10 was necessary due to the requirements set forth by the school administrators.

**Warwick-Edinburgh Mental Well-being Scale (WEMWS).**   The Warwick-Edinburgh Mental Well-being Scale (WEMWBS) was employed to complement the assessment of mood states by capturing positive emotional experiences. The WEMWBS is a self-administered questionnaire comprising 14 items aimed at evaluating an individual's overall well-being [28]. The scale is designed to offer a comprehensive measurement of well-being, encompassing affective-emotional, cognitive-evaluative, and psychological functioning dimensions. Participants rated their happiness on a 5-point scale from "never" to "always" based on their actual feelings in the past two weeks according to the guideline, with higher scores indicating higher levels of experienced well-being. The Chinese version of the scale has been used preliminarily and has shown good reliability and validity [29].

In the present study, the WEMWBS was employed to assess and compare mood experiences between athletes and students in Sample 2 and Sample 3. The substitution of the POMS with the WEMWBS was necessary due to the requirements set forth by the school administrators.

## Procedure

**Process of epidemic prevention covered in this study.**   The 2022 Shanghai Omicron wave emerged in late February and expanded rapidly [30]. To accelerate the suppression of the epidemic's spread, the districts of Shanghai began to temporarily closed-off in phases at the end of March.

Beginning in mid-April, all districts launched classified management of city areas (the number of infection numbers reported in a single day reached a peak in mid-April). With the community transmission of the virus under control, the city gradually restored production and daily life in early June.

*Closed-loop management.* In response to the Omicron wave, athletes and most coaches who train at the Shanghai Chongming Sports Training Base were placed under closed-loop management starting from March 7, 2022. Subsequently, administrators, scientists, and medical teams involved in the training of sports teams were also placed under closed-loop management by March 14, 2022. Furthermore, logistics services personnel essential to keep the operation of the training base running were included in the closed-loop management on March 22. The release from the closed-loop management was aligned with the progress of daily routine revival in Shanghai, with personnel at the training base being allowed to apply for release from closed-loop management in early June. On June 17, the close-loop management on all personnel, coaches, and athletes was entirely lifted.

*Management in the community.* During the epidemic control process, the majority of high school students adhered to the home study requirement and remained isolated at home. Community management imposed strict restrictions on residents, limiting their mobility to prevent the spread of the virus. Consequently, the vast majority of residents, including high school students, were subjected to a stringent "no leave home unless necessary" policy for an extended duration. In Shanghai, high school students ceased offline classes on 12 March and subsequently transitioned to online lessons that followed the same schedule as the former, developed by the municipal education department. These lessons, including physical education classes, required students to attend classes via television or computer from Monday to Friday, starting from morning to late afternoon. They were also expected to interact with their teachers online, complete their daily homework, and undertake regular school-organized exams through the internet. Furthermore, physical activities that were part of the physical education classes had to be performed at home.

**Procedure of measurement.** The first test (PSQI and POMS) started in the fourth week after the athlete was closed-loop managed as a 1-month closed-loop management status (end of March). The sports teams involved in this study organized their athletes to complete the test independently over a two-week period. This period was caused by a few sports teams' delayed confirmation of participation in the study. Some districts were locked down at the start of the first test, and by the end of the first test, all districts were locked down (0–2 weeks).

After every district had been in lockdown for at least four weeks, the scales (PSQI, PSS-10, and WEMWBS) were distributed to students in selected high school classes, according to the high school's schedule, and students were given one week to complete the scales. Some of the scales were replaced in response to suggestions from school administrators. When testing begins, students have been studying at home for 7 weeks (the first 2–3 weeks are not all home isolated). Districts have been under lockdown for 4–5 weeks at this point.

The second test (PSQI, POMS, PSS-10, and WEMWBS) was started in mid-May, and the athletes finished it in 2 weeks as the 2-month closed-loop management status. The training base entered the 10th week of closed-loop management (about 6 weeks after the final athlete completed the first test) at the start of the second test. Districts have been under lockdown for 6–7 weeks at this point.

All scales were completed online. The sports team administrators or school administrators, respectively, distribute the link of the scales and remind athletes or students to complete it timely.

## Statistics

Statistical analysis was performed using jamovi Version 2.3 [31]. Paired-sample *t*-tests were conducted to examine differences in the indicators of sleep quality and mood among athletes during the closed-loop management period of one month and two months (Sample 1).

**Table 1. Descriptive results of sleep quality of athletes in closed-loop management ($M \pm SD$).**

|  | 1-month | 2-month | $p$ | Cohen's d |
|---|---|---|---|---|
| PSQI | 6.96 ± 2.77 | 7.01 ± 3.58 | 0.873 | -0.016 |
| Subjectively rated sleep quality | 1.19 ± 0.64 | 1.26 ± 0.76 | 0.250 | -0.099 |
| Sleep time (minute) | 1351.31 ± 42.53 | 1355.26 ± 48.76 | 0.269 | -0.086 |
| Wake time (minute) | 433.84 ± 32.49 | 427.18 ± 32.51 | 0.002 | 0.205 |
| Sleep latency (minute) | 28.65 ± 20.93 | 30.36 ± 23.89 | 0.357 | -0.076 |
| Hours of sleep (hour) | 7.35 ± 0.73 | 7.17 ± 0.94 | 0.024 | 0.217 |
| Sleep efficiency (%) | 0.83 ± 0.12 | 0.84 ± 0.12 | 0.452 | -0.072 |
| Sleep disorder | 6.14 ± 4.44 | 6.62 ± 4.88 | 0.178 | -0.103 |
| Daytime dysfunction | 3.29 ± 1.59 | 2.95 ± 1.68 | 0.022 | 0.212 |

Independent samples $t$-tests were employed to compare the indicators of sleep quality and mood between adolescent athletes (Sample 2) and high school students (Sample 3).

## Results

### 1-month vs. 2-month closed-loop management

Results of sleep showed no difference in PSQI score, $t(109) = -0.16$, $p = 0.873$. Subjectively rated sleep quality did not differ, $t(109) = -1.16$, $p = .250$. Sleep time did not differ, $t(109) = -1.11$, $p = 0.269$. The result of wake time revealed a difference, $t(109) = 3.19$, $p = 0.002$, Cohen's d = 0.205. wake-up time was earlier at 2-month than at 1-month (see Table 1). There was no difference in sleep latency, $t(109) = -0.92$, $p = 0.357$. There was a significant difference in sleep hours, $t(109) = 2.29$, $p = 0.024$, Cohen's d = 0.217. Hours of sleep was shorter at 2-month than at 1-month. Sleep efficiency was not different, $t(109) = -0.76$, $p = 0.452$. Sleep disorder did not show a difference, $t(109) = -1.36$, $p = 0.178$. There was a difference in daytime dysfunction, $t(109) = 2.32$, $p = 0.022$, Cohen's d = 0.212. Daytime dysfunction at 2-month was less severe than at 1-month.

Results of mood revealed no difference in tension, $t(109) = -0.94$, $p = 0.348$. There was a difference in anger, $t(109) = -2.49$, $p = 0.014$, and Cohen's d = -0.231. Anger was more sever at 2-month than at 1-month (see Table 2). There was no difference in fatigue, $t(109) = 0.55$, $p = 0.583$. Depression was similarly unaffected, $t(109) = -1.77$, $p = 0.080$. In terms of vigor, there was no change, $t(109) = 0.52$, $p = 0.601$. The difference in confusion was not significant, $t(109) = -1.08$, $p = 0.282$. The difference in esteem was similarly not significant, $t(109) < 0.001$, $p > 0.999$.

**Table 2. Descriptive results of mood of athletes in closed-loop management ($M \pm SD$).**

|  | 1-month | 2-month | $p$ | Cohen's d |
|---|---|---|---|---|
| Tension | 5.88 ± 4.29 | 6.25 ± 4.70 | 0.348 | -0.090 |
| Anger | 5.16 ± 4.78 | 6.37 ± 5.72 | 0.014 | -0.237 |
| Fatigue | 8.24 ± 4.63 | 8.03 ± 3.95 | 0.583 | 0.053 |
| Depression | 4.96 ± 4.91 | 5.83 ± 5.41 | 0.08 | -0.167 |
| Vigor | 10.58 ± 4.39 | 10.38 ± 4.57 | 0.601 | 0.050 |
| Confusion | 4.76 ± 3.49 | 5.09 ± 3.85 | 0.282 | -0.103 |
| Esteem | 6.83 ± 3.21 | 6.83 ± 3.26 | > 0.999 | <0.001 |

**Table 3. Descriptive results of sleep quality of athletes and students ($M \pm SD$).**

| | Athletes | Students | *p* | Cohen's d |
|---|---|---|---|---|
| PSQI | 6.49 ± 3.14 | 4.66 ± 2.84 | <0.001 | 0.626 |
| Subjectively rated sleep quality | 1.22 ± 0.70 | 0.76 ± 0.72 | <0.001 | 0.634 |
| Sleep time (minute) | 1361.70 ± 42.38 | 1407.96 ± 55.81 | <0.001 | -0.887 |
| Wake time (minute) | 422.71 ± 30.88 | 430.25 ± 27.10 | 0.065 | -0.267 |
| Sleep latency (minute) | 25.86 ± 18.37 | 15.74 ± 20.60 | <0.001 | 0.507 |
| Hours of sleep (hour) | 7.28 ± 0.79 | 7.06 ± 1.05 | 0.114 | 0.229 |
| Sleep efficiency (%) | 0.87 ± 0.09 | 0.91 ± 0.10 | 0.006 | -0.402 |
| Sleep disorder | 6.03 ± 4.09 | 2.51 ± 3.02 | <0.001 | 1.043 |
| Daytime dysfunction | 2.94 ± 1.75 | 2.61 ± 1.91 | 0.220 | 0.177 |

### Closed-loop management vs. management in the community

Results of sleep revealed a difference in PSQI scores, $t(227) = 4.35$, $p < 0.001$, Cohen's d = 0.626. Athletes had worse sleep quality problems than students (see Table 3). Subjectively rated sleep quality differed, $t(227) = 4.41$, $p < 0.001$, Cohen's d = 0.634. Athletes rated their sleep quality problems worse than students. Differences was also found in sleep time, $t(227) = -6.16$, $p < 0.001$, Cohen's d = -0.887. Athletes slept earlier than students. Wake time was not different, $t(227) = -1.85$, $p = 0.065$. Differences existed in sleep latency, $t(227) = 3.52$, $p < 0.001$, Cohen's d = 0.507. Athletes had a longer latency than students. There was no difference in hours of sleep, $t(227) = 1.59$, $p = 0.114$. A difference existed in sleep efficiency, $t(227) = -2.79$, $p = 0.006$, Cohen's d = -0.402. Athletes slept less efficiently than students. Differences existed in sleep disturbance, $t(227) = 7.24$, $p < 0.001$, Cohen's d = 1.043. Athletes had more serious sleep disorder than students. Daytime dysfunction was not different, $t(227) = 1.23$, $p = 0.220$.

The results for mood showed differences in perceived distress, $t(227) = -2.91$, $p = 0.004$, Cohen's d = -0.419. Athletes perceived lower distress than students (see Table 4). There was a difference in belief about coping capacity, $t(227) = -2.24$, $p = 0.026$, Cohen's d = -0.322. Coping capacity were lower in athletes than in students. There was no difference in positive mood experience, $t(227) = -1.62$, $p = 0.107$.

## Discussion

The results revealed no significant change in sleep quality as the length of closed-loop management increased, but a significant reduction in sleep duration and a reduction in daytime dysfunction. In terms of mood, only anger occurred significantly elevated. When compared to the students studying at home, the athletes' sleep quality appeared significantly worse. Athletes, in particular, experienced significant deficiencies in subjective sleep quality, sleep latency, sleep efficiency, and sleep disorder. These features, however, did not change across closed-loop

**Table 4. Descriptive results of mood of athletes and students ($M \pm SD$).**

| | | Athletes | Students | *p* | Cohen's d |
|---|---|---|---|---|---|
| PSS-10 | Perceived stress | 14.36 ± 4.43 | 16.25 ± 4.53 | 0.004 | -0.419 |
| | Coping capacity | 11.15 ± 3.40 | 12.25 ± 3.44 | 0.026 | -0.322 |
| WEMWBS | | 46.01 ± 11.31 | 48.64 ± 11.22 | 0.107 | -0.233 |

management, suggesting that they are stable negative characteristics of athletes. In terms of mood, athletes reported lower stress levels but also lower confidence in dealing with stress.

Several large-sample surveys of athletes have revealed that lockdown has negative effects such as decreased sleep efficiency, increased sleep duration, delayed sleep, and difficulty falling asleep [7,8,32]. In contrast to these findings, athletes reported shorter sleep duration but lower sleep problem behaviors in an attempt to "quarantine" camp training [21]. The current study found that, similar to the changes observed in this practice, sleep duration decreased significantly as time passed under closed-loop management. In addition, contrary to previous research that found lockdown to cause athletes to sleep late and wake up late [32], we discovered that this trend may be associated with an earlier waking time rather than a later falling asleep time. This finding adds to previous research by examining changes in longer duration and more detailed sleep indicators [21]. Furthermore, while athletes' sleep duration was reduced and the mean values obtained are close to the suggested lower bound of sleep duration for athletes [33], daytime dysfunction decreased with prolonged closed-loop management. This finding could imply that athletes recovered more quickly from fatigue by sleeping for shorter periods of time. Overall, sleep quality has remained stable, but some positive changes may have occurred.

The logistical support and regulations of closed-loop management may be the primary contributors to the phenomenon observed in this study. Previous research has shown that in a closed-loop management setting, sleep behavior problems can be reduced [21]. Athletes in closed-loop management have almost no opportunity to go out (for vacation or competition) and thus can receive scientifically proven training, dietary, and rehabilitation programs more systematically, and the effects of these scientific training tools may gradually manifest themselves over time, even in their sleep, with closed-loop management. More importantly, closed-loop management could have aided in the improvement of sleep hygiene. This is especially important for non-clinical sleep problems [34]. This closed-loop management eliminated some sleep-harming habits, such as nighttime entertainment and dining out, as well as nighttime dorm parties. Furthermore, sports team administrators were present throughout the closed-loop management period, allowing for effective practice and flexible adjustment of sleep hygiene regulations.

Although studies on "quarantine" training revealed that athletes had better sleep quality when compared to the home isolation phase [21,22], no studies have compared the differences between closed-loop management in athletes and lockdown management in the general population in the same period. For the first time, this was investigated using athletes and high school students of the same age group. The comparison results revealed that, while athletes' sleep quality did not change significantly during closed-loop management, they had more significant sleep quality issues throughout the procedure than students. It is important to note that sleep quality problems were relatively common among athletes even before the epidemic [35–37]. Given that the primary purpose of closed-loop management is to allow athletes to return to regular training [21], these problems with sleep quality may show exactly the characteristics of the athlete's sleep in general.

The results of the current study were different from those of previous research in terms of mood. A study that tracked adolescent athletes for 10 weeks during a lockdown observed a decrease in depression and confusion [18]. The current study, however, discovered that only anger seems to have significantly increased during closed-loop management. During the first lockdown following the pandemic outbreak, one study found that Polish athletes had higher levels of anger than Spanish athletes, raising the possibility that cultural differences and the degree of agreement with the management approach may be the cause of this discrepancy [38]. Another study has also found that negative emotions, such as anger, were reduced after

the lockdown was lifted and enhanced after re-entry into semi-closure from the perspective of a prolonged lockdown [39]. More referentially, Shenzhen, China, experienced a week-long lockdown at the end of March. A study on this lockdown found a negative correlation between residents' anger and confidence in anti-epidemic efforts [40]. The closed-loop management in the current study was extended over time, and no one knew how long it would last. Such uncertainty may exacerbate anger by influencing psychosocial factors such as confidence and trust [40].

When compared to the lockdown that the general public received, closed-loop management also has unique properties in terms of its impact on mood. Studies have found that closed management will lead to increased negative emotions such as stress [14]. It has also been discovered that athletes are more anxious than non-athletes during lockdown [41]. However, athletes in the current study had lower levels of perceived stress than their peers of the same age. This finding is consistent with previous research comparing the stress levels of athletes participating in "bubble" training camps before lockdown, during the lockdown, and during the camp, which found that the majority of athletes, including Malaysian-Chinese athletes, perceived lower levels of stress during the camp [21,22]. This study once again demonstrated the significance of closed-loop management. The athletes' ability to maintain relatively normal training and life in closed-loop management may be the primary cause of this effect.

Based on the findings of this study, the following aspects need to be taken into consideration when adopting closed-loop management in the future so that athletes' sleep and emotional state during closed-loop management can be guaranteed. In terms of sleep, it is first important to note that the sleep quality problems that were already present in the athlete population remain salient in closed-loop management. This asks for the integration of sleep quality monitoring and intervention in the sports team's support services during closed-loop management. In the process of closed-loop management, athletes may lose sleep due to early wakeups if the period of closure is prolonged. This problem can be addressed by adding naps after the closure to make up for lost sleep time [42], or by delivering sleep hygiene instruction at the start of the lockdown. In terms of mood, it is necessary to pay attention to the timely explanation and persuasion of management rules and policies in the process of closed-loop management as a way to reduce the possibility of athletes developing negative emotions over time. Additional education and training of athletes' stress coping abilities are also required to improve their self-efficacy in dealing with stress induced by the epidemic [19]. It should not be forgotten, however, that the above recommendations are based on the findings of the study on longer-term closed-loop management. Based on the previous studies' recommendations, shorter closed-loop management cycles should be considered first, as conditions permitted [22].

This study has several limitations that should be considered. First, the lack of baseline data limits the interpretation of the results. In this study, closed-loop management was extended over time (more than 3 months) to meet the requirement of citywide epidemic prevention, resulting in a late study design and implementation. As a result, the optimal time to test baseline levels was missed. Despite this limitation, the study focused on comparing the effects of the duration of closed-loop management and the differences between closed-loop management and management in the community. While some new findings emerged, the lack of baseline data remains a limitation. Second, the study only analyzed the impact of approximately 2 months of closed-loop management and did not analyze the impact of a longer period of closed-loop management. It is important to acknowledge that this study was conducted in parallel with the whole city epidemic control process and not actively recruited and operated by the researcher, which limited the duration of the study. Nevertheless, the

study's results may to some extent represent the impact of longer closed-loop management and can inform decision-makers for future epidemic control. Third, data from high school students were used to compare the management of closure with closures in the community. Although the students included in the study were similar to athletes in terms of lifestyle, and the school's physical education include a portion of specialized sports training, their differences in certain physical or psychological aspects of their characteristics from professional athletes might mask some of the differences caused by the management approach. This was a compromise in the management of epidemic control due to the reluctance to obtain a sample under epidemic control, and should be taken into account when interpreting the results. Finally, the content and methods of measurement did not include athletes' sleep behavior and objective sleep indicators. This was mainly due to the fact that the athlete sleep behavior questionnaire was not revised in Chinese beforehand and could not be applied in the current study. Additionally, the lack of objective measurement equipment available on the training base during closed-loop management, such as actigraphy, and the consideration of hygiene issues in cross-use of the limited number of devices, did not allow for the use of commonly used objective measures. To sum up, the characteristics of sleep and mood states of athletes who received closed-loop management during the Shanghai 2022 Omicron pandemic were examined in this study. In terms of sleep, the prolonged duration of the closure resulted in athletes waking up earlier, resulting in less sleep time. During the same time period, athletes' sleep quality was worse than that of the general population in the community. In terms of emotions, the prolonged closure time enhanced athletes' anger, whereas athletes felt less stressed than the general population in the community. Closed-loop management had an impact on athletes' sleep and mood in both positive and negative ways. The findings of this study hold significant implications for the development of preplanning strategies aimed at implementing closed-loop management for athletes engaged in competitive sports, thereby augmenting the protective effects of such measures on athletes. Beyond its relevance for competitive sports training, this study may also have practical implications for the design of preplanning strategies for other critical industries and municipalities operating under emergency conditions, thereby enhancing its societal impact.

## Supporting information

**S1 Data.**
(RAR)

## Author Contributions

**Conceptualization:** Chenhao Tan, Jinhao Wang.

**Data curation:** Chenhao Tan, Jinhao Wang.

**Formal analysis:** Chenhao Tan.

**Funding acquisition:** Jun Qiu.

**Investigation:** Chenhao Tan, Jun Yin, Guohuan Cao, Lu Cao, Chao Chen.

**Methodology:** Chenhao Tan, Chao Chen.

**Project administration:** Jun Qiu.

**Resources:** Jun Yin, Lu Cao, Chao Chen.

**Supervision:** Jinhao Wang.

**Writing – original draft:** Chenhao Tan.

**Writing – review & editing:** Chenhao Tan, Jinhao Wang, Guohuan Cao, Lu Cao, Jun Qiu.

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
