## [Decision Letter · Decision Letter 0]

6 Mar 2023

PONE-D-23-01195The effect of prolonged closed-loop management on athletes' sleep and mood during COVID-19 pandemic: Evidence from the 2022 Shanghai Omicron wavePLOS ONE

Dear Dr. Qiu,

Thank you for submitting your manuscript to PLOS ONE. After careful consideration, we feel that it has merit but does not fully meet PLOS ONE’s publication criteria as it currently stands. Therefore, we invite you to submit a revised version of the manuscript that addresses the points raised during the review process.

We look forward to receiving your revised manuscript.

Kind regards,

Md Maruf Ahmed Molla

Academic Editor

PLOS ONE

Journal Requirements:

Additional Editor Comments:

This is a well written paper. Please revise the paper according to our reviewers' comments

Reviewers' comments:

Reviewer's Responses to Questions

**Comments to the Author**

1. Is the manuscript technically sound, and do the data support the conclusions?

Reviewer #1: Partly

Reviewer #2: Yes

Reviewer #3: Yes

2. Has the statistical analysis been performed appropriately and rigorously? 

Reviewer #1: Yes

Reviewer #2: Yes

Reviewer #3: Yes

3. Have the authors made all data underlying the findings in their manuscript fully available?

Reviewer #1: Yes

Reviewer #2: Yes

Reviewer #3: Yes

4. Is the manuscript presented in an intelligible fashion and written in standard English?

Reviewer #1: Yes

Reviewer #2: Yes

Reviewer #3: Yes

5. Review Comments to the Author

Reviewer #1: 1.Materials and methods

Participants: Sample 1 consisted of athletes who were in closed loop since the omicron wave and sample 2 consisted of athletes who were in closed loop for two months. The time interval may not have been significant enough to justify the findings.

Comparing Levels of stress and anger in the general population who were in lockdown ( Sample 3) to that of athletes in closed loop management seemed unnecessary as in closed loop only regular training can be done but in both cases the activities of daily life are hampered . So justification required.

As this study was conducted over a period of more than 2 months it is only natural that some participants withdrew their informed consent. So elaboration required on that aspect.

Why are sample 2 and 3 merged?? So what is the benefit of taking sample 2 ??

Sleep medication aspect is taken into consideration in athletes but not in general population

Before June 17 th, we’re any personnel in the closed loop allowed to leave?? Elaboration required

In general population the school going children have a different biophysical profile to that of athletes of the same age . This is not taken into consideration as this can be a confounding factor in not only sleep quality, duration but also on the mood and perceived stress on the participants

2. Recommendations

As the study conducted only during 1 wave among a small sample size are the results generalizable?

Recommendations are as per the objectives, but a separate subsection may be done

Reviewer #2: Dear author,

Congratulation for your excellent study. I find your paper very information and interesting. Below are some of my comments.

Abstract:

Suggest to include the statistical analysis employed in the study.

Introduction:

Informative, relevant and arranged accordingly.

Data sources - is there exact congruence between data of athlete (from routine psy services for sport team) with data of students (from school mental health education prog)?

Methodology:

What test family (t test?) and statistical test did the author employ for G*Power computation of sample size?

Measurement is precise and well-elaborated.

Statistical analysis employed is accurate and appropriate with study objectives.

Results are presented clearly and tabulation format is relevant with the statistical analysis.

Discussions are congruent with study objectives and results. References are up-to-date.

Reviewer #3: This manuscript highlights the importance of sleep and sleep studies in the psychological well-being of athletes compared to the general population. The study does conform to the format of a standard research study and presentation. The relevance of the paper goes without saying. The paper details the limitations of the study which are mainly due to the timing during the COVID-19 pandemic, however, in my opinion, it’s the COViD-19 pandemic that makes the findings of this study relevant and critical in the life, training, and vocation of professional athletes. COVID-19 basically assists in emphasizing the effects of prolonged close-loop management on an athlete’s sleep and mood during pandemics requiring quarantine and isolation due to its mode of transmission and control.

The objectives of the study, and the method, has significant limitations as alluded to in the paper, but he results and discussions do lead to the conclusions and recommendations.

The indicators applied for the assessment namely the Pittsburgh Sleep Quality Index, the Profile Mode States, the Perceptual Stress Scale, and Warwick-Edinburg Mental Well-being should have received a more general description in the paper to improve its public or general understanding and appeal. How these studies resulted in the simple-term outputs and outcomes should also be provided in simple terms. I hope I’m clear over here.

Inasmuch as the manuscript assists in closing certain knowledge gaps in science, its audience seems narrow and lacks general appeal.

The manuscript may be published with or without addressing the minor comments made as it does make a significant contribution to science and knowledge with empirical data.

Thanks for the opportunity to review the manuscript.

6. PLOS authors have the option to publish the peer review history of their article (what does this mean?). If published, this will include your full peer review and any attached files.

Reviewer #1: **Yes: **Dr Lina Bandyopadhyay

Reviewer #2: **Yes: **Hafizuddin Bin Awang

Reviewer #3: **Yes: **Dr. Nana-Kwadwo Biritwum

---

## [Author Response · Author response to Decision Letter 0]

28 Mar 2023

Reviewer #1: 1.Materials and methods

• Participants: Sample 1 consisted of athletes who were in closed loop since the omicron wave and sample 2 consisted of athletes who were in closed loop for two months. The time interval may not have been significant enough to justify the findings.

Response: Thanks a lot for this question! We may not have been clearly expressed in the manuscript about the samples and time intervals. We apologize for any confusion and would like to explain this below. First of all, regarding the sample, sample 1 were athletes who had been in closed management at the training base for 2 months and had completed two scale-based tests approximately 1 month and 2 months after the training base closure. Sample 2 were 16-18 years old adolescent athletes in closed-loop management for comparison with students of the same age.

For the interval, previous studies have analyzed the mood and sleep status of athletes during a one-month 'quarantine' camp and demonstrated the positive effects of this type of management. This study expands on the contribution of that study by analyzing the effect of prolonged closed-loop management on athletes during a real-life city lockdown in 2022. As this study was conducted during a citywide lockdown at the same time, the duration of closed-loop management was determined by the government administration, so the duration of closed-loop management could not be controlled by the researcher. Although there may be shortcomings in the duration, the differences found in this study provide further evidence-based information on closed-loop management as a strategy for epidemic management and add to previous research.

We have updated these in the Introduction, Methods and Discussion sections.

Introduction: “Closed-loop management has emerged as a promising approach to strike a balance between epidemic prevention/control and training effectiveness. This study seeks to expand on previous research regarding the effects of one month of closed-loop management on sleep and mood among athletes. Specifically, the study aims to compare the impact of closed-loop management over a longer duration (approximately 2 months) both longitudinally (in comparison to the effects observed after approximately 1 month) and horizontally (in comparison to management in the community / home isolation) to comprehensively evaluate the efficacy of this management style. To achieve these objectives, the study undertook an analysis of the sleep and mood of professional athletes under closed-loop management at different time points (after approximately 1 month and after 2 months) during the Omicron Wave in Shanghai, China, 2022. Additionally, the study conducted a cross-sectional comparison between high school-aged athletes under closed-loop management and high school students of the same age who were in home isolation for approximately 2 months. High school students were chosen due to their requirement to follow a daily schedule of online classes, including physical education, while in isolation, thereby offering a comparable routine to that of the athletes. The study's findings have practical implications for promoting targeted sleep and mood precautions during future outbreaks or other public health emergencies.”

Methods: “Sample 1 was used to compare the athletes' sleep and mood after 1 month and 2 months of closed-loop management. It comprised 249 athletes aged 16 years and above who trained at Shanghai Chongming Sports Training Base during the Omicron wave in Shanghai, China, 2022. All athletes were subjected to closed-loop management. The sample size represented the vast majority of athletes in the targeted age range who trained at the facility under closed-loop management during the study period. However, not all athletes participated in both tests due to decisions made by administrators of each sports team on participation in each test. Specifically, 178 athletes participated in the test after 1 month of closure, while 181 athletes participated in the test after 2 months. Among these athletes, 110 participants (70 females) took part in both tests, providing longitudinal data for analysis. The average age of the participants was 19.42 years (SD = 2.99). The sports included in the study were fencing, modern pentathlon, field hockey, Chinese martial arts, gymnastics, boxing, badminton, judo, taekwondo, and karate.

In this study, obtaining permission from sports team administrators was necessary for athletes to participate and for the use of their data. However, some athletes did not participate in either the first or second test due to sports team administrators failing to provide timely feedback on their intention to participate, or because they did not perceive the need for their sports team to participate. Notably, none of the athletes themselves expressed any desire to withdraw from the study.

Sample 2 and Sample 3 were used for comparison between closed-loop management and management in the community (home isolation). 

 Sample 2 consisted of 69 athletes aged 16-18 years who participated in the test after 2 months of closure management (43 females), with a mean age of 16.96 years (SD = 0.83). The sports included fencing, modern pentathlon, field hockey, handball, Chinese martial arts, gymnastics, boxing, badminton, judo, taekwondo, and basketball.

Sample 3 consisted of 160 high school students (85 females), with a mean age of 16.10 years (SD = 0.30). The students came from a senior high school in Shanghai. The students were required to attend online classes and interact with their teachers for the majority of the day, from Monday to Friday. They were also required to complete homework after class and participate in physical education classes that were included in the curriculum and needed to be completed at home. The schedule was set by the city, and the online classes were recorded and broadcast by the city. This daily routine is similar to that of the athlete's training and study routine in the training base. The students who were isolated at home did not have to deal directly with many of the material concerns of life, similar to the athletes who were training in the training base. None of the participants in Sample 2 had any prior experience of being placed in isolation in a cubicle hospital or isolation site. At the time of testing, all students had been studying at home for approximately 2 months.”

Discussion: “…Second, the study only analyzed the impact of approximately 2 months of closed-loop management and did not analyze the impact of a longer period of closed-loop management. It is important to acknowledge that this study was conducted in parallel with the whole city epidemic control process and not actively recruited and operated by the researcher, which limited the duration of the study. Nevertheless, the study's results may to some extent represent the impact of longer closed-loop management and can inform decision-makers for future epidemic control….”

•Comparing Levels of stress and anger in the general population who were in lockdown ( Sample 3) to that of athletes in closed loop management seemed unnecessary as in closed loop only regular training can be done but in both cases the activities of daily life are hampered . So justification required.

Response: Thank you for raising this important point! We apologize for the lack of clarity in our methods section. While both groups faced limitations in their daily lives, our study focuses on the impact of prolonged closed-loop management. In order to assess this impact, we deemed it important to examine both longitudinal, temporal changes (i.e., changes over 1 month and 2 months) and horizontal, differences in approach (i.e., differences between closed-loop management and general home isolation). Our objective in comparing athletes in closed-loop management to the general population in closed management is to compare the effects of the extended duration of closed-loop management longitudinally, while also comparing the effects of the two management methods horizontally over the same period of time (i.e., close to or more than 2 months). This approach allows us to demonstrate the effects of closed-loop management after a longer period of impact. At the time of the study, the general population was subject to a strict "no leave home unless necessary" policy, similar to the short-term isolation of athletes returning from competing abroad, while the closed-loop athletes were able to maintain a comparable level of specialized training at the training base, with less restriction on movement. This comparison between the two approaches is of interest and may provide valuable insight for future management practices. We have addressed this section in the introduction and methods section to provide additional clarification.

Introduction: “Closed-loop management has emerged as a promising approach to strike a balance between epidemic prevention/control and training effectiveness. This study seeks to expand on previous research regarding the effects of one month of closed-loop management on sleep and mood among athletes. Specifically, the study aims to compare the impact of closed-loop management over a longer duration (approximately 2 months) both longitudinally (in comparison to the effects observed after approximately 1 month) and horizontally (in comparison to management in the community / home isolation) to comprehensively evaluate the efficacy of this management style. To achieve these objectives, the study undertook an analysis of the sleep and mood of professional athletes under closed-loop management at different time points (after approximately 1 month and after 2 months) during the Omicron Wave in Shanghai, China, 2022. Additionally, the study conducted a cross-sectional comparison between high school-aged athletes under closed-loop management and high school students of the same age who were in home isolation for approximately 2 months. High school students were chosen due to their requirement to follow a daily schedule of online classes, including physical education, while in isolation, thereby offering a comparable routine to that of the athletes. The study's findings have practical implications for promoting targeted sleep and mood precautions during future outbreaks or other public health emergencies.”

Methods: “Sample 2 and Sample 3 were used for comparison between closed-loop management and management in the community (home isolation). 

 Sample 2 consisted of 69 athletes aged 16-18 years who participated in the test after 2 months of closure management (43 females), with a mean age of 16.96 years (SD = 0.83). The sports included fencing, modern pentathlon, field hockey, handball, Chinese martial arts, gymnastics, boxing, badminton, judo, taekwondo, and basketball.

Sample 3 consisted of 160 high school students (85 females), with a mean age of 16.10 years (SD = 0.30). The students came from a senior high school in Shanghai. The students were required to attend online classes and interact with their teachers for the majority of the day, from Monday to Friday. They were also required to complete homework after class and participate in physical education classes that were included in the curriculum and needed to be completed at home. The schedule was set by the city, and the online classes were recorded and broadcast by the city. This daily routine is similar to that of the athlete's training and study routine in the training base. The students who were isolated at home did not have to deal directly with many of the material concerns of life, similar to the athletes who were training in the training base. None of the participants in Sample 2 had any prior experience of being placed in isolation in a cubicle hospital or isolation site. At the time of testing, all students had been studying at home for approximately 2 months. ”

“Management in the community: During the epidemic control process, the majority of high school students adhered to the home study requirement and remained isolated at home. Community management imposed strict restrictions on residents, limiting their mobility to prevent the spread of the virus. Consequently, the vast majority of residents, including high school students, were subjected to a stringent "no leave home unless necessary" policy for an extended duration. In Shanghai, high school students ceased offline classes on 12 March and subsequently transitioned to online lessons that followed the same schedule as the former, developed by the municipal education department. These lessons, including physical education classes, required students to attend classes via television or computer from Monday to Friday, starting from morning to late afternoon. They were also expected to interact with their teachers online, complete their daily homework, and undertake regular school-organized exams through the internet. Furthermore, physical activities that were part of the physical education classes had to be performed at home.”

•As this study was conducted over a period of more than 2 months it is only natural that some participants withdrew their informed consent. So elaboration required on that aspect.

Response: Thank you for your important question regarding our methodology. We acknowledge that participant dropout did occur during the study. This research was conducted as part of the daily sport team support process, and informed consent was obtained from both team administrators and athletes prior to each test, including permission for their data to be used in the study. The main reason for participant dropout in our study was that some sports team administrators did not provide timely feedback on participation or did not perceive participation as necessary, which resulted in some athletes not participating in the test. It should be noted that the athletes did not actively refuse to participate. In response to this issue, we have made revisions to the methods section of the manuscript to provide more details on participant dropout.

Methods: “In this study, obtaining permission from sports team administrators was necessary for athletes to participate and for the use of their data. However, some athletes did not participate in either the first or second test due to sports team administrators failing to provide timely feedback on their intention to participate, or because they did not perceive the need for their sports team to participate. Notably, none of the athletes themselves expressed any desire to withdraw from the study.”

•Why are sample 2 and 3 merged?? So what is the benefit of taking sample 2 ??

Response: Thank you for your question, and we apologize for any confusion that may have been caused by the statement in the text. The reference to "sample 2 and 3 merged" in the methods section was intended to describe the use of two separate samples to compare the differences between the two methods of isolation (closed-loop management and home isolation), rather than to combine the two samples into one. We have made the necessary correction to the text to clarify this point.

Methods: “Sample 2 and Sample 3 were used for comparison between closed-loop management and management in the community (home isolation). ”

•Sleep medication aspect is taken into consideration in athletes but not in general population

Response: Thank you for bringing up this question! We did not include sleep medicine (medication use) as a dependent variable for consideration in either athletes or the general population in our study. There are a few reasons for this. Firstly, for athletes, anti-doping regulations may require strict diagnosis and drug exemptions to be granted for sleep medication use, and athletes generally do not choose to take such medication. Furthermore, during the lockdown period, it is difficult for athletes to seek medical attention for sleep problems, as the entire city is shut down due to epidemic control measures. Secondly, for students, the school psychology teachers have information on medical visits and diagnoses involving sleep medication and psychotropic substances, and there is specific management of such students. However, no information on medication use was obtained from the teacher in this study. Lastly, access to medical care and the purchase of medication (which must be medically prescribed) is very difficult during the lockdown period, which leads to the fact that sleep medication treatment in general can be largely disregarded. Therefore, we felt that including sleep medication as an intervention (i.e., the medication dimension of the PSQI) would interfere with the judgement of outcomes, as they are essentially constrained by a combination of management and external conditions. We have added more information about this in the methods section.

Methods: “In regards to sleep medication, Chinese athletes are required to obtain administrative approval for use exemptions under anti-doping regulations after receiving medical advice. Similarly, for students, the use of psychotropic substances is subject to information collection in schools for youth protection purposes, allowing for school psychologists to intervene and provide assistance when necessary. However, in this study, neither type of formal use of sleep medication was found. Furthermore, the scarcity of medical resources during the epidemic control period made seeking medical attention for sleep problems highly unlikely. As such, changes in the use of sleep medication can be largely disregarded in this study. To avoid confounding the results with factors limiting medication use, the dimension of sleep medication was not included in the statistical analysis.”

•Before June 17 th, we’re any personnel in the closed loop allowed to leave?? Elaboration required

Response: Thank you for this question. We apologize for the lack of clarity regarding the significance of the date in question. After the implementation of citywide epidemic control measures, the city began to gradually lift the lockdown starting on June 1st. The training base closure was also gradually lifted, and by June 17th it was essentially lifted entirely. We have updated the methodology section to better convey this information.

Methods: “In response to the Omicron wave, athletes and most coaches who train at the Shanghai Chongming Sports Training Base were placed under closed-loop management starting from March 7, 2022. Subsequently, administrators, scientists, and medical teams involved in the training of sports teams were also placed under closed-loop management by March 14, 2022. Furthermore, logistics services personnel essential to keep the operation of the training base running were included in the closed-loop management on March 22. The release from the closed-loop management was aligned with the progress of daily routine revival in Shanghai, with personnel at the training base being allowed to apply for release from closed-loop management in early June. On June 17, the close-loop management on all personnel, coaches, and athletes was entirely lifted.”

•In general population the school going children have a different biophysical profile to that of athletes of the same age . This is not taken into consideration as this can be a confounding factor in not only sleep quality, duration but also on the mood and perceived stress on the participants

Response: Thank you for this good question, we really appreciate it. In our study, we made a deliberate decision to compare students with athletes of the same age in order to provide a more comprehensive analysis of the effects of extended closed-loop management (beyond 2 months) on athletes in a cross-sectional manner by comparing closed-loop management with home isolation.

The rationale for choosing students as a comparison group is that during the closure period, school students are required to attend online classes according to a daily schedule, including physical activity classes completed at home. Similarly, athletes are also required to attend different training and academic sessions according to their daily schedule, leading to relatively consistent work and rest schedules between the two groups.

While we acknowledge that there may be differences in the physical and psychological characteristics of athletes and students, we attempted to match the groups as closely as possible by age and sport. Our sample of Shanghai students are in the school that has integrated sports-specific training with physical education, and the school has also formed sports teams under professional coaches to train and participate in competitions. We have included this information in the introduction to the sample.

Despite our efforts to control for possible confounding factors, we recognize that differences between students and professional athletes still exist. This was a trade-off decision due to the limited availability of home isolated athletes during our study. While we did recruit a small sample of home isolated chess players, they were not representative enough, and their form of sport may not be transferable to other sports, particularly in terms of sports physiology. Ultimately, we decided to use student data as a comparison group equally with athletes of the same age group, with possible confounding factors added to the limitations section.

Methods: “The students were required to attend online classes and interact with their teachers for the majority of the day, from Monday to Friday. They were also required to complete homework after class and participate in physical education classes that were included in the curriculum and needed to be completed at home. The schedule was set by the city, and the online classes were recorded and broadcast by the city. This daily routine is similar to that of the athlete's training and study routine in the training base. The students who were isolated at home did not have to deal directly with many of the material concerns of life, similar to the athletes who were training in the training base. None of the participants in Sample 2 had any prior experience of being placed in isolation in a cubicle hospital or isolation site.”

Discussion: “Third, data from high school students were used to compare the management of closure with closures in the community. Although the students included in the study were similar to athletes in terms of lifestyle, and the school's physical education include a portion of specialized sports training, their differences in certain physical or psychological aspects of their characteristics from professional athletes might mask some of the differences caused by the management approach. This was a compromise in the management of epidemic control due to the reluctance to obtain a sample under epidemic control, and should be taken into account when interpreting the results.”

2. Recommendations

•As the study conducted only during 1 wave among a small sample size are the results generalizable?

•Recommendations are as per the objectives, but a separate subsection may be done

Response: Thank you for your suggestion! We appreciate the opportunity to clarify the sample size for this study. Sample 1 included a total of 249 professional athletes aged 16 and over from various sports, including fencing, modern pentathlon, hockey, martial arts, gymnastics, boxing, badminton, judo, taekwondo, and karate. This sample size represents the vast majority of age-eligible athletes residing at the training base and encompasses a broad range of sports. However, due to the decision of sports team administrators, some athletes only participated in the first test while others only participated in the second test, resulting in a total of 110 athletes who completed both tests in full. We have updated the text to include a clear description of the sample size.

Methods: “It comprised 249 athletes aged 16 years and above who trained at Shanghai Chongming Sports Training Base during the Omicron wave in Shanghai, China, 2022. All athletes were subjected to closed-loop management. The sample size represented the vast majority of athletes in the targeted age range who trained at the facility under closed-loop management during the study period. However, not all athletes participated in both tests due to decisions made by administrators of each sports team on participation in each test. Specifically, 178 athletes participated in the test after 1 month of closure, while 181 athletes participated in the test after 2 months. Among these athletes, 110 participants (70 females) took part in both tests, providing longitudinal data for analysis.”

Reviewer #2: Dear author,

Congratulation for your excellent study. I find your paper very information and interesting. Below are some of my comments.

Abstract

Suggest to include the statistical analysis employed in the study.

Response：Many thanks for this suggestion. We have updated the abstract.

Abstract: “Paired sample t-tests and independent sample t-tests were used for comparisons across different time intervals and different management approaches.”

Introduction:

Informative, relevant and arranged accordingly.

Data sources - is there exact congruence between data of athlete (from routine psy services for sport team) with data of students (from school mental health education prog)?

Response：Thank you for raising this question. The psychological tests utilized in the school's mental health education program were designed collaboratively by the school's counselors, administrators, and members of our research team, after receiving informed consent for their involvement in this study. The data analyzed in this study were collected specifically for this research. Subsequent to the data collection, the school's psychological counselor provided some follow-up services, including targeted online lectures, based on the results of the data analysis. It is important to note that the content of these services is not directly related to the present study.

Methods: “This study used data from the psychological services undertaken by the Institute during the lockdown. Data of athletes were obtained from routine psychological services for sports teams. Data of the students were obtained through the school's psychological health education program implemented during the epidemic. The test used in this program was adapted from tests conducted in sports teams, with some minor adjustments made in accordance with the school administration's requirements. Following the tests, the school's psychology teachers provided mental health education based on the results.”

Methodology:

What test family (t test?) and statistical test did the author employ for G*Power computation of sample size?

Response：Thank you for raising this question. We have updated the method section to provide more information regarding the statistical analysis conducted in this study. Specifically, we utilized a paired sample t-test to analyze sample 1 and an independent sample t-test to analyze samples 2 and 3. 

Methods: “Paired-sample t-tests were conducted to examine differences in the indicators of sleep quality and mood among athletes during the closed-loop management period of one month and two months (Sample 1). Independent samples t-tests were employed to compare the indicators of sleep quality and mood between adolescent athletes (Sample 2) and high school students (Sample 3).”

Measurement is precise and well-elaborated.

Statistical analysis employed is accurate and appropriate with study objectives.

Results are presented clearly and tabulation format is relevant with the statistical analysis.

Discussions are congruent with study objectives and results. References are up-to-date.

Reviewer #3: This manuscript highlights the importance of sleep and sleep studies in the psychological well-being of athletes compared to the general population. The study does conform to the format of a standard research study and presentation. The relevance of the paper goes without saying. The paper details the limitations of the study which are mainly due to the timing during the COVID-19 pandemic, however, in my opinion, it’s the COViD-19 pandemic that makes the findings of this study relevant and critical in the life, training, and vocation of professional athletes. COVID-19 basically assists in emphasizing the effects of prolonged close-loop management on an athlete’s sleep and mood during pandemics requiring quarantine and isolation due to its mode of transmission and control.

The objectives of the study, and the method, has significant limitations as alluded to in the paper, but he results and discussions do lead to the conclusions and recommendations.

Response：Thank you for acknowledging our study! We conducted our research in parallel with a city-wide epidemic control process, which involved a significant degree of uncertainty. However, despite the challenging circumstances, we strived to carry out data collection and comparison that was both accurate and valid to the best of our ability, given the limited resources and conditions that were available during the lockdown. Our goal was to provide valuable insights that could inform future closed-loop management efforts.

The indicators applied for the assessment namely the Pittsburgh Sleep Quality Index, the Profile Mode States, the Perceptual Stress Scale, and Warwick-Edinburg Mental Well-being should have received a more general description in the paper to improve its public or general understanding and appeal. How these studies resulted in the simple-term outputs and outcomes should also be provided in simple terms. I hope I’m clear over here.

Response: Thank you for your question and your helpful suggestions. We have revised our manuscript to better describe the scale and its dimensions. 

Methods: “(1) global PSQI score, the total score (0-21) of PSQI, with higher scores indicating poorer sleep quality overall; (2) subjectively rated sleep quality, score on the subjective sleep quality dimension of the PSQI (0-3), with higher scores indicating worse subjective perceived sleep quality; (3) Sleep time, the time when participant go to bed (in minutes since 0:00,if later than 24:00, 1440 minutes are superimposed), the bigger the value, the later the bedtime; (4) Wake time, the time participant wake up after going to bed (in minutes since 0:00), the greater the value, the later the morning wake time; (5) Sleep latency, the time between going to bed and falling asleep, with larger values indicating that it takes more time to fall asleep; (6) hours of sleep, the total number of hours slept, with higher values indicating longer sleep duration; (7) calculated sleep efficiency, the proportion of total sleep time actually spent asleep after falling asleep, with higher values indicating higher sleep efficiency; (8) total sleep disorder score, the total score for the sleep disorder dimension of the PSQI (0-30), with higher scores indicating more severe sleep disorder symptoms (symptoms such as difficulty falling asleep, waking up at night, excessive dreaming, etc.); and (9) total daytime dysfunction score, the total score for the daytime dysfunction dimension of the PSQI (0-6), with higher values indicating more pronounced daytime dysfunction (feelings like sleepiness, problem in keep up enough enthusiasm, etc.). ”

“tension (e.g. restless, nervous), anger (e.g. peeved, bitter), fatigue (e.g. weary, bushed), depression (e.g. hopeless, helpless), vigor (e.g. cheerful, energetic), confusion (e.g. bewildered, forgetful), and esteem (e.g. proud, satisfied)”

“The PSS-10 is made up of six negative items (perceived distress, the lack of control and negative reactions) and four positive items (coping capacity, the degree of ability to cope with existing stressors)”

“The WEMWBS is a self-administered questionnaire comprising 14 items aimed at evaluating an individual's overall well-being. (27). The scale is designed to offer a comprehensive measurement of well-being, encompassing affective-emotional, cognitive-evaluative, and psychological functioning dimensions.”

Inasmuch as the manuscript assists in closing certain knowledge gaps in science, its audience seems narrow and lacks general appeal.

Response: We are glad that you found our study to be valuable. Our findings can have significant positive implications for professional athletes, sports clubs, and sports administration in their efforts to prepare for and manage any future health crises using closed-loop management. Moreover, during the period of closure, several other essential businesses and institutions in the city, including manufacturing companies, also utilized closed-loop management strategies. Thus, we believe that the implications of our study can extend beyond the sports industry to these industries as well, providing useful insights for designing preplans for health emergencies and other related aspects. We have accordingly made appropriate modifications to the implications section of our manuscript to reflect these broader applications.

Discussion: “The findings of this study hold significant implications for the development of preplanning strategies aimed at implementing closed-loop management for athletes engaged in competitive sports, thereby augmenting the protective effects of such measures on athletes. Beyond its relevance for competitive sports training, this study may also have practical implications for the design of preplanning strategies for other critical industries and municipalities operating under emergency conditions, thereby enhancing its societal impact.”

The manuscript may be published with or without addressing the minor comments made as it does make a significant contribution to science and knowledge with empirical data.

Thanks for the opportunity to review the manuscript.

Response：Thank you very much for your positive and very helpful advice!

PLOS authors have the option to publish the peer review history of their article (what does this mean?). If published, this will include your full peer review and any attached files.

No

---

## [Editor Report · Decision Letter 1]

10 Apr 2023

The effect of prolonged closed-loop management on athletes' sleep and mood during COVID-19 pandemic: Evidence from the 2022 Shanghai Omicron wave

PONE-D-23-01195R1

Dear Dr. Qiu,

We’re pleased to inform you that your manuscript has been judged scientifically suitable for publication and will be formally accepted for publication once it meets all outstanding technical requirements.

Kind regards,

Md Maruf Ahmed Molla

Academic Editor

PLOS ONE

Additional Editor Comments (optional):

Thank you for your revised paper. We believe its ready for acceptance now. 
---

## [Editor Report · Acceptance letter]

13 Apr 2023

PONE-D-23-01195R1 

The Effect of Prolonged Closed-loop Management on Athletes' Sleep and Mood during COVID-19 Pandemic: Evidence from the 2022 Shanghai Omicron Wave 

Dear Dr. Qiu:

I'm pleased to inform you that your manuscript has been deemed suitable for publication in PLOS ONE. Congratulations! Your manuscript is now with our production department. 

Kind regards, 

on behalf of

Dr. Md Maruf Ahmed Molla 

Academic Editor

PLOS ONE